# Type of Findings Generated by the Occupational Therapy Workforce Research Worldwide: Scoping Review and Content Analysis

**DOI:** 10.3390/ijerph19095307

**Published:** 2022-04-27

**Authors:** Tiago S. Jesus, Karthik Mani, Claudia von Zweck, Sureshkumar Kamalakannan, Sutanuka Bhattacharjya, Ritchard Ledgerd

**Affiliations:** 1Center for Education in Health Sciences, Institute for Public Health and Medicine, Feinberg School of Medicine, Northwestern University, Chicago, IL 60611, USA; 2Department of Occupational Therapy, School of Health Professions, The University of Texas Medical Branch at Galveston, Galveston, TX 77555, USA; kamani@utmb.edu; 3World Federation of Occupational Therapists (WFOT), 1211 Geneva, Switzerland; opservices@wfot.org (C.v.Z.); executivedirector@wfot.org (R.L.); 4Department of Social Work, Education and Community Well-Being, Northumbria University, Newcastle upon Tyne NE7 7XA, UK; sureshkumar.kamalakannan@northumbria.ac.uk; 5Department of Occupational Therapy, Byrdine F. Lewis College of Nursing and Health Professions, Georgia State University, Atlanta, GA 30302, USA; sbhattacharjya@gsu.edu

**Keywords:** health workforce, health personnel, human resources for health, occupational therapists, rehabilitation, review

## Abstract

Occupational therapists are needed to meet the health and occupational needs of the global population, but we know little about the type of findings generated by occupational therapy workforce research conducted worldwide. We aim to synthesize these findings and their range of content to inform future investigations. A scoping review with content analysis was used. Six scientific databases, websites of official institutions, snowballing, and key informants were used for searches. Two independent reviewers took selection decisions against the eligibility criteria published a priori in the review protocol. Of the 1246 unique references detected, 57 papers were included for the last 25 years. A total of 18 papers addressed issues of attractiveness and retention, often in Australia, and 14 addressed the issues of supply, demand, and distribution, often in the US. Only these two categories generated subtopics. Many workforce issues were rarely addressed as a main topic (e.g., race/ethnic representation). Cross-national, cross-regional, or cross-professional studies generated more actionable findings. Overall, we found few discernable trends, minimal evidence of research programs, and various gaps in content coverage or in the use of contemporary research approaches. There is a need for a coordinated strengthening of the occupational therapy workforce research worldwide.

## 1. Introduction

Occupational therapists are health professionals needed to meet the health, rehabilitation, and occupational needs of people experiencing a wide range of health conditions and disabilities [1,2]. Occupational therapists also address well-being and health promotion [3], as well as human rights and occupational injustices arising from social determinants of health [4,5,6]. To fulfill their societal role, occupational therapists need to be in sufficient supply, equitably distributed across geographic and practice areas, motivated in their work, and in compliance with competency standards [1]. 

Research on the health workforce in general [7,8,9,10,11] and on the occupational therapy workforce in particular [1,12,13,14,15] can help assess these requirements are met, find ways by which these requirements can be improved, and provide the evidence base to inform and evaluate population-centered workforce policies and practices [7,9,16]. However, little is known about the global status of the occupational therapy workforce research, both in scope and findings type.

To inform global directions for the occupational therapy workforce research, the World Federation of Occupational Therapists (WFOT) initiated a process to review the occupational therapy workforce research worldwide [17]. In a first paper, the review has identified quantitative trends of the occupational therapy workforce research and found minimal growth year over year (i.e., 14 years for an additional yearly publication), an over-reliance on cross-sectional study designs, little use of inferential statistics, and a minority of papers (25–30%) reporting funding support and more advanced study methods [18]. The study question for this paper is: what are the types of findings generated by occupational therapy workforce research conducted worldwide? Overall, we aim to synthesize these types of findings as one means to inform future investigations. 

## 2. Materials and Methods

A scoping review was developed. Scoping reviews often address exploratory research questions on broad topics to identify key concepts, research methods, or types of evidence for detecting gaps in an area of investigation [19,20,21,22]. We followed the Arksey and O’Malley’s framework [19,20,23] and the recent Joanna Briggs Institute’s guidance for conducting scoping reviews [24]. The review protocol has been peer-reviewed and published [17].

### 2.1. Searches

Six electronic databases of the scientific literature (Medline/PubMed, Web of Science, Scopus, CINAHL, PDQ-Evidence; OTseeker) were systematically searched. The protocol details the search strategy for PubMed database [17], used to guide the searches in the other databases. The detailed search strategy for each database is outlined in the Appendix A. The search strategy was appraised against The Peer Review of Electronic Search Strategies guidelines [25]. The searches were conducted in June 2021.

The grey literature and official research-based reports were identified through screening and keyword searches in international institutional websites: WFOT; World Health Organization; Health Workforce Research (European Public Health Association); and international websites focused on the occupational therapy profession [17]. Finally, we used snowballing (e.g., references of included papers) as well as key informants (i.e., representatives of WFOT member organizations) to find additional references. 

### 2.2. Eligibility Criteria

We included occupational therapy workforce research fitting at least one category of workforce research defined in the study protocol (see Table 1) [17]. The design of the inclusion categories was informed by a WFOT position statement [1], a review of the rehabilitation workforce literature [13], the global strategy on the health workforce [26], and a reader of health workforce research [9].

Studies regarding the education of occupational therapists from a curriculum or pedagogical perspective, as well as occupational health investigations, were excluded. Methodologically, we included any quantitative, qualitative, or mixed-methods research, case studies, and systematic reviews published in peer-reviewed journals or in official institutional venues. No language restrictions were applied [17], but only papers reported in English (or simultaneously in English and another language for journals publishing articles in more than one language) were identified. We excluded commentaries, letters, posters, study protocols, and broad databases or papers without a study question, replicable methods, or interpretation of the results.

Papers on the occupational therapy workforce or with occupational therapists as participants were included, even if other workers were studied, although either comparative or stratified results for occupational therapists were required. 

With these criteria, two independent reviewers (TJ, KM) conducted titles-and-abstracts screening and full-text reviews, after attaining an 80% or greater agreement in pilot tests on at least 5% of the references. Up to two discussion rounds among the reviewers were used for consensus on the final eligibility. 

We reviewed the literature from all geographic areas with no restrictions on time elapsed since publication; subsequently, a cut-off at saturation level was decided upon and it was collectively determined to analyze only papers published in the last 25 years [17]. 

### 2.3. Data Extraction

We extracted the methodological features (e.g., study design, participants), geographic areas or settings, the key findings, either quantitative or qualitative, and any stated implications. After a pilot test with 10% of the included references, one experienced reviewer (TJ) extracted the information, fully verified by another research author (either KM, SK, or SB). As common in scoping reviews [27,28], quality appraisals were not performed.

### 2.4. Data Synthesis

We synthesized the key findings from the literature based on an inductive content analysis [29], with category themes emerging from the findings. As we included literature from a wide range of topics, methods, geographic areas, and dates, a configurative, narrative-based reporting of each category is provided. 

## 3. Results

Figure 1 provides the Preferred Reporting Items for Systematic Reviews and Meta-Analyses (PRISMA) flowchart of this review. Out of 1226 unique references, 57 papers were included after the temporal cut-off, i.e., published in the last 25 years.

Organized by emergent categories (headings) and subcategories (subheadings), key findings are outlined below, beginning with the most common. Only two categories generated subcategories.

### 3.1. Attractiveness and Retention

This category was addressed by 18 papers, under the subtopics below.

#### 3.1.1. Reasons to Choose and Remain in a Career

A survey study in Israel found that occupational therapists more heavily weighed the importance of professional and academic stature, while speech and language pathologists (SLPs) demonstrated preferences for terms of employment and benefits, as well as work conveniences [30]. A similar study in the US found that occupational therapists highly rated the importance of professional growth, an environment aligned with their values, adequate support staff, and diversity of practice, whereas SLPs valued other factors; these findings informed the use of profession-specific tailored retention strategies [31].

#### 3.1.2. Determinants of Attrition

In a recent survey study in Sweden, factors underlying workforce attrition were investigated among the 35% of occupational therapists who in a general workforce showed the intention to leave the profession [32]. Frequent determinants related to the need to improve the work environment were stress and high work pressure, perception of unfair wages, organizational dysfunction, limited managerial support, or poor recognition of knowledge and work [32]. In a Korean study, structural equation modelling determined that presenteeism (i.e., continuing to work while sick or exhausted) played a statistically significant mediating role between occupational stress and an intention to leave the profession; conversely, the perception of occupational therapists of organizational support served to buffer turnover intention [33].

Among older studies, a survey of job satisfaction among occupational therapists in Australia found that 60% of the respondents intended to leave the profession within 10 years, with a lack of promotion opportunities serving as a leading factor of dissatisfaction [34]. Within the context of being a minority gender within the Canadian occupational therapy profession, 74% of male occupational therapists in Canada reported that they would leave the profession within 10 years [35]. 

#### 3.1.3. Intention to Leave Mental Health Practice 

In a large, metropolitan healthcare organization delivering mental health care in Australia, a series of studies on burnout, job satisfaction, and intention to leave the profession was conducted [35,36,37,38]. Based on multivariate analyses [36,37,38,39] results showed that occupational therapists working in mental health reported low to moderate turnover intention. Receiving good remuneration, attaining recognition, and being challenged at work were associated with higher job satisfaction; those who experienced difficulties with managers had lower satisfaction [38]. Thirty-three percent of variation relating to intention to leave was predicted by job satisfaction, which was strongly associated with recognition and work/life balance [37]. Occupational therapists (*n* = 34 out of 277 participants) had the greatest intention to leave their job among the five professional groups studied [36]. 

#### 3.1.4. Attractiveness of Mental Health as a Practice Area

In a study of the attractiveness of mental health practice across eight countries, common pull forces for leaving a position included a lack of resources, a poor work environment, and low recognition of the occupational therapy role in meeting mental health needs. Among country differences, in both Canada and South Africa, more than 70% of the respondents perceived that it was “not easy to fill” vacancies in mental health settings, in contrast to 16% in Sweden [40].

A pilot experiment in Australia to evaluate novel fieldwork placement models in mental health found that students involved (*n* = 6) reported they were likely to work in mental health care upon graduation [41]. In a systematic review of the value of continuing professional development to enhance recruitment and retention of occupational therapists in mental health, a total of 13 articles were identified but the evidence linking continuing professional development and recruitment and retention was neither strong nor conclusive [42].

#### 3.1.5. Retention in Rural Areas

In rural areas of an Australian state, 75% of the occupational therapists reported having had a career interruption, the highest value among all allied health professionals, while the average of all the occupations combined was 57% [43]. In turn, a review of factors influencing occupational therapy students’ perceptions of rural practices found that positive fieldwork experiences promoted rural career intentions, despite the lack of a formal specialization in rural practice [44]. 

A qualitative study of those who left rural practice in Australia reported that participants identified the lack of access to professional development as a significant factor in their decision to leave [45]. Also in Australia, a survey of occupational therapy private practitioners in rural areas found that one third had other paid employment. In addition, one third believed they could not further sustain their practice, as only face-to-face contact was reimbursed and the population served had lower income levels or health insurance coverage [46].

For an Australian university educating occupational therapists for rural practice, a survey study found that occupational therapy students’ perceptions towards rural practice changed over the course of their studies for considering rural and remote practice (*p* = 0.003). Influential factors reported included good fieldwork experience (*p* = 0.039) and inspiring fieldwork supervisors (*p* = 0.01). The curriculum and a student’s rural background were not found to be significant factors [47]. 

### 3.2. Supply, Demand, and Distribution

This category was addressed by 14 papers, under the subtopics below.

#### 3.2.1. Supply and Demand, including Forecasts

In the US, the supply of occupational therapists forecasted for 2030 was expected to increase by 45% from 104,290 in 2016. Demand was anticipated to rise by 22% under two different scenarios, suggesting a sufficient supply to meet the projected demand [48]. However, another study using different indicators and methods found that demand for occupational therapy services was expected to far outpace the supply of occupational therapists within the US in the subsequent 20 years, with many states facing supply shortages [49]. 

Using vacancies to determine demand, a US study identified a national vacancy rate of 8.9% for occupational therapists, with rates up to 11.9% for the western region. Sixty-seven percent of employers reported difficulty hiring occupational therapists, with more than half of the vacant positions unfilled for 6 months or more [50]. Previously, at a regional level, 24% of the employers reported occupational therapy vacancies and 63% had difficulties hiring occupational therapists [51]. The authors concluded that sustained workforce shortages result in increased workload and less time for non-reimbursable job-related activities. Therapy assistants were also reported to be in short supply, confounding occupational therapy workforce issues [50]. On the supply side, the WFOT routinely monitors the occupational therapy workforce worldwide based on data collected by member associations, which have shown the diversity of the profession’s demography between countries and provides comparative data for an occupational therapy workforce planning [15].

#### 3.2.2. Supply and Distribution per Country Region and Sector

A study across US counties found that the absolute number of occupational therapists increased more in the areas already better supplied than those with official shortages [52]. In an earlier systematic search analysis of diverse data sources in Australia, little information on occupational therapy workforce supply and distribution was identified, hampering workforce planning [53].

A survey undertaken in a Canadian province determined that short-staffing, limited work flexibility, and lack of acknowledgement or career advancement were deterrents in public practice for those who preferred to switch to private practice [12]. For the rural areas of an Australian state, the private–public practice ratio of occupational therapists was reported as 0.35:1, compared to 0.75:1 for the overall allied health professionals [43]. In Brazil, a government-based database was used to determine that 1323 occupational therapists were officially registered as working in public services of the social sector [54].

A comprehensive description of changing demographic trends in South African occupational therapy indicated that the population-adjusted ratio of occupational therapists grew from 0.4 to 0.9 per 10,000 population over 10 years after 2009, with a simultaneous decrease of registered occupational therapy technicians and occupational therapy assistants [55]. Occupational therapists were primarily located in densely populated, urbanized provinces, with up to five times the difference in population-adjusted ratios. Only 25% of occupational therapists were employed by the public health service [55]. 

#### 3.2.3. Supply, Utilization, and Unmet Demand by Service Type

A recent US study focused on hospice care found that only 10.6% of patients received occupational therapy, even though a large number (87.3%) had functional limitations in the activities of daily living [56]. Also in the US, a systematic review of the workforce literature for autism-related child healthcare services identified unmet occupational therapy needs that varied between 14% to 46%, depending on geography and timeframe [57]. A Netherlands study on the availability of allied health care in nursing homes determined the availability of 0.9 full-time equivalents per 100 beds for occupational therapy compared to 2.16 for physical therapy [58]. In Switzerland, a recent study of workforce supply in primary care practices found only three occupational therapists available, compared to 78 physical therapists [59]. 

### 3.3. Staffing Levels—Upscaling and Monitoring

Five papers addressed this category.

In Saudi Arabia, local epidemiological data were combined with international staffing recommendations, across professions, to estimate future staff requirements and their costs. The study concluded that 137.1 full-time equivalents in occupational therapy were required to meet national staff requirements; occupational therapy had the greatest need across all professions, with no further internists, physiotherapists, or dietitians required [60].

When comparing staffing levels for stroke rehabilitation against guidelines in England’s public hospitals, 84% of units reached the guideline for occupational therapy, more than for other therapy professions (42% and 16%); however, a post-hoc analysis of therapy staffing guidelines revealed incongruencies that threatened the validity of the results [61].

In the US, two recent, statistically advanced studies examined changes in therapy staffing in skilled nursing facilities after a new payment model was implemented to disincentivize intense therapy provision. Both studies showed significant staffing reductions affecting therapists and assistant therapists. One study found reductions of 6.1% for occupational therapists compared to 5.5% for physical therapists and 9–10% for therapist assistants [62]. The second study identified an immediate decline of 5.2% for both occupational therapists and physical therapists, and a 7.6% decline for assistant therapists [63].

Finally, a recent study in Norway in the context of an administrative reform in municipalities found that new occupational therapy positions were most frequently established in medium-sized municipalities that already had occupational therapists. The number of prior positions and being in the process of merging with another municipality were the only statistically significant predictors for establishing new occupational therapy positions [64]. 

### 3.4. Competencies, Regulation, and Continuous Education Requirements

Five papers addressed this category.

In a recent Australian study, a stepwise focus group and Delphi process were used to revise and validate the competency standards for occupational therapy driver assessors. All statements achieved consensus with an average consensus at 96.8%, resulting in the development of revised standards endorsed by Occupational Therapy Australia [65]. In an earlier study, the overall Australian Occupational Therapy Competency Standards were subject to a multi-stakeholder evaluation and consultation process that enabled a set of recommendations and revisions to be developed with national support and ownership [66]. 

A documentary analysis in the US compared state regulatory board requirements for continuous professional development and found substantial variation for licensure renewal, ranging from no requirements to renewal cycles from 1 to 3 years, with continuing education hours required per renewal cycle from 10 to 40 h [67]. The same study also highlighted the scant use of an existing national template for assigning continuing education credits; therefore, most state boards were missing specific information or had ambiguous requirements [67]. Furthermore, no difference was found for licensees with different job responsibilities such as clinicians, academicians, or administrators/managers, with several regulatory boards requiring continuing education related to patient care delivery regardless of the licensee’s responsibilities [67]. Some years before, a British survey study found that 31% of the occupational therapists surveyed were not aware of the national registration body’s recommendations on continuous professional education [68]. 

Finally, a survey study with 546 occupational therapists was conducted in Canada, after the occupational therapy regulatory organization in the province of Quebec implemented a compulsory ePortfolio for continuous professional development. Having a clear understanding of the ePortfolio goals and objectives was found to positively influence its acceptability and perceived impact [69].

### 3.5. Human Resources Management—Recruitment, Leadership, and Workload

Four papers addressed this category.

A study of employer recruitment strategies in the Canadian provinces of Alberta and Saskatchewan showed that the most effective approaches in 2005 were word of mouth, advertising in general media, providing student fieldwork placements, offering varying financial incentives, and an emergent focus on website job postings [70]. In an earlier study in the UK to explore the factors attracting newly qualified occupational therapists to job advertisements, the British Journal of Occupational Therapy was found to be both a frequent and effective source of recruitment advertising for both employers and prospective employees at that time [71].

A recent survey study on the leadership styles amongst occupational therapy clinicians in Australia found leadership scores close to the 40th percentile, consistently lower than those reported for geographically relevant norms; better outcomes were associated with a transformative leadership style [72].

Finally, regarding workload prediction for time management and staffing determinations, a recent systematic review found no workload prediction models for occupational therapists, which contrasts for example with the nursing profession [73]. The authors concluded that occupational therapist time (highly individualized for each client) may not be as easily categorized into specific tasks, time to complete them, or in a time-use or workload formula as seen in the nursing literature [73]. In addition, the volume of research and understanding of the variables impacting time-use has been stronger for the nursing profession [73].

### 3.6. International Mobility—And Integration of Those Internationally Trained

Three papers addressed this category.

A secondary analysis in the US combining diverse sources of workforce and immigration data found that 8% of the occupational therapists are foreign-born and entered the US at age 21 or older. This figure was considered the best available estimate for the number of US internationally trained occupational therapists, lower than the 8% of pharmacists, 12% of lab technicians, and 12% of physical therapists. The authors concluded that a source of accurate data on international mobility in occupational therapy was not available [74].

In turn, a recent qualitative study in Canada focused on the support and integration of internationally-trained occupational therapists among the 30 that registered for a practice preparation course. Only 13 completed the compulsory courses, with reasons for withdrawal including insufficient time and questioning of program relevance [75]. Overall, participants needed more than exam preparation and may have underestimated their learning needs related to their transition to Canadian practice; individualized learning needs assessments and education might be therefore required [75]. These implications aligned with a previous qualitative study also in Canada that identified the need to address a workforce-integration continuum from coming to Canada to integrating into the workplace [76].

### 3.7. Career Transitions—To Supervisor Roles, Return to the Career 

Two papers addressed this category.

A qualitative study at one metropolitan healthcare organization in Australia aimed to identify struggles and uncover insights for support strategies on the transition of occupational therapist to senior career, supervisory roles. From the insights (e.g., related to readiness and preparedness for progression, the unknowns and the expectations of what it takes), the research team implemented new initiatives to smooth the transition to higher-level responsibilities such as early career opportunities for staff to participate in activities similar to those required at a supervisory level (e.g., co-convene some department-based committees or quality-improvement processes) [77].

Finally, a qualitative study in New Zealand explored how occupational therapists managed a return to practice after a career break. Maintaining a sense of professional connectedness and professional identity before and during a career break was found to enable a successful transition back into professional practice [78]. 

### 3.8. Salary Levels 

Two papers addressed this category. 

In India, a survey of various levels of workforce data for occupational therapists found statistically significant associations in the gender–wage gap favoring males (*p* = 0.0012) and for higher salary ranges for those with a master or doctorate degrees (OR: 3.03, 95% CI: 1.57–5.88). Oddly, those with clinical job titles had greater salary ranges relative to those in academia or management positions (OR: 2.84; 95% CI: 1.32–6.12) [79]. This finding can reflect a growing supply–demand gap specifically for clinical practitioners, but bonuses, allowances, leave, and flexibility in hours were not accounted for and can be better for those in non-clinical roles [79]. Finally, a recent study in the US for skilled nursing facilities found that the wage ratio of occupational therapists and occupational therapist assistants was 1.4 on average and equivalent to the ratio for physical therapists and physical therapist assistants [80].

### 3.9. Training of New Graduates 

Two papers addressed this category.

An analysis of undergraduate programs in Brazil from 1991 to 2008 for several health professionals found that in 2008 the admissions for occupational therapy students were 1620 but with only 26% of the vacancies filled, the second lowest among 12 professions, compared to 8552 for physical therapy, i.e., 5.3 times the difference in the public education sector, but 30 times the difference in the private education sector [81].

For anglophone Sub-Saharan Africa, a recent study found that only 16 occupational therapy training programs were identified, all at or below the bachelor’s level except for South Africa. None of the post-secondary educational institutions offering degree programs in occupational therapy were found in Botswana, Eritrea, Gambia, Lesotho, Liberia, Malawi, Namibia, Seychelles, Sierra Leone, Swaziland, and Zambia [82].

### 3.10. Racial/Ethnic Representation

Only one paper addressed this category.

A recent study published in the *JAMA* regarding racial/ethnic representation in the US healthcare workforce found that occupational therapists had the second highest percentage of Whites (81%) among 10 healthcare professions (e.g., physicians had 62%). As the representation of Black graduates was lower than in the current workforce (e.g., occupational therapy: 0.31 vs. 0.50), the current workforce trend would not be reversed [83].

## 4. Discussion

This scoping review has identified ten major topics in the occupational therapy workforce research worldwide. The issues of attractiveness and retention (18 papers), mostly in Australia, and supply, demand, and distribution (14 papers), primarily in the US, were common topics. These categories generated subtopics such as attractiveness and retention in rural areas or mental health practices, and the supply and distribution per region or sector. No other topic was addressed by more than five papers or generated subtopics. Several patterns of findings in this review as well as topics that were minimally or not addressed are worth discussing.

Although attractiveness and retention were relatively common themes, the findings were essentially descriptive and exploratory; a lack of more advanced studies is evident, such as the discrete choice experiments informing recruitment or retention packages as seen elsewhere in the health workforce research [84,85,86]. In turn, experimental research findings were absent across topics; hence, even in the more frequently addressed topics, we know little about what interventions result in improved attractiveness, retention and job satisfaction, or reduced turnover.

As seen in other areas of health workforce research [87,88], cross-national or cross-regional, cross-professional, or benchmark comparisons provided more insightful information (i.e., grounded on both similarities and differences) than merely profiling, exploring, or describing variables for the occupational therapy workforce alone. For example, occupational therapists were found to value opportunities for professional growth and professional recognition more than other professions. This information can help to inform profession-specific human resource strategies, in addition to factors already known to be associated with better health workforce retention overall [89,90]. 

Supply, demand, and service utilization were the most common topics studied in the US, including forecasting workforce needs. Interestingly, two forecasting approaches in the US, with varying methods and indicators, led to opposite conclusions regarding shortages or lack thereof for the occupational therapy workforce. Occupational therapists can work in several practice areas of the healthcare sector, in communities for health promotion or occupational justice, as well as in the educational, labor, vocational, and social sectors. Employer types may vary highly across nations and include unique sectors, e.g., municipalities in Scandinavian countries where occupational therapist per population ratios are amongst the highest worldwide [15,32,40]. Determining demand for occupational therapists therefore may require a mix of data regarding population need (e.g., epidemiological indicators [14,91]) in addition to a sector-by-sector or service-by-service demand analysis within a country or jurisdiction. For example, the study in Saudi Arabia included in this review [60] found the need to scale up the occupational therapy workforce, although the same conclusions were not found for other professions. The sum of such detailed analyses, by services or sectors, can be more precise and account for distribution issues when compared with a country-wide, whole-sectors estimation of occupational therapy workforce needs. The findings of this review did not reveal any regional, country, or cross-country situational analysis (i.e., data-based, system-wide, stakeholders-inclusive analysis and deliberation toward planning) of the occupational therapy workforce, alone or alongside other rehabilitation professions, despite such studies being common elsewhere [92]. 

We found no comprehensive study of the labor market for occupational therapy. For example, investigations have been conducted in the nursing profession that addressed broad-ranging issues including underemployment across public or private sectors [93,94]. Similarly, no studies of performance-enhancing strategies (e.g., health worker supervision approaches) were identified [95,96]. Although the WFOT monitors the occupational therapy workforce worldwide [15], no comprehensive global reports were found regarding estimates of shortages, scope of practice analyses, policy response frameworks, exemplars of successful developments, or economic analyses (e.g., return-of-investment analysis) [97,98,99]. While investment in human resources for health typically strengthens the health system as well as generating employment and contributing to economic growth [26], the only economic analysis found was related to the cost of the scale-up of the stroke rehabilitation workforce in Saudi Arabia that included occupational therapists [60].

Possible actionable results were found regarding the variation of licensing requirements across states within the same country, i.e., the US [67]. This finding came from the single documentary analysis included in this review. Documentary and health policy analysis is often used in workforce research [100,101] and cross-national comparisons of variations on regulation across jurisdictions are common in health workforce research [102,103], but were underused in the reviewed literature. The systematic review found no use of workload models and indicators [73], while these are commonly utilized for the continuous advancement of the nursing and other health professions [104,105,106,107].

Sequences of studies (i.e., research programs) were rarely used, perhaps with the exception of a study on job satisfaction and turnover intention in a mental health service in Australia. The investigations progressed from an exploratory analysis to more advanced correlations and cross-professional comparisons [36,38,39], but still without experimental approaches. Transition to supervisory roles, return after career break, and equitable racial/ethnic representation (in which occupational therapists were amongst the least diverse workforce) were examples of topics not addressed by more than single papers. Studies on international mobility and salary issues also were rarely addressed, and only a few research-based processes to develop competency standards were found. In short, apart from attraction and retention in Australia and supply and demand in the US, there is no discernible trend in the content of the occupational therapy workforce research worldwide.

We found no explicit focus or findings on task shifting and task sharing processes [108], although one paper in South Africa discussed the issue [55]. Task shifting or task sharing strategies can help supply population needs especially, but not exclusively, in low-resource and underserved settings. Many African countries do not offer post-secondary programs for educating occupational therapists; a lack of research capacity may therefore partly explain the lack of investigations in this area [82]. In addition, a low occupational therapy workforce supply limits the higher-level tasks that can be shifted to occupational therapists (e.g., prescription of assistive devices for official or reimbursement purposes) and the lower-level tasks that could be partly shifted, shared, and supervised by occupational therapists and delivered by middle- or lower-level cadres. 

Finally, many of the workforce research results need constant updating. Unlike many clinical or biological studies with results that can remain valid for a long time, workforce research findings are responsive to political, legal, geographic, economic, sociological, or cultural changes over time. For example, given the rise in new communication technologies, the findings of studies related to the recruitment of occupational therapists that were conducted more than a decade ago are not necessarily applicable today. Push and pull factors for issues such as private practice can also vary over time because of labor market dynamics, reimbursement, economic crises, or any other major societal change [109]. Such requirements for constant updating reinforce the need for continuous workforce research programs that were found to be absent in the occupational therapy literature.

### Study Limitations

The results synthesized in this paper are illustrative of the type of findings the occupational therapy workforce research reported. As this is a scoping review, which does not apply quality appraisals, the finding reported here should be interpreted with caution. Our synthesis focused on reporting key findings and immediate implications from what the papers reported in the context of their study design, context, and timing. As a result, although the findings were clustered into content categories, we did not aggregate or combine findings from different studies. Although we have used a combination of databases searches with other search strategies (e.g., snowballing, grey literature searches, key-informant recommendations), one cannot be sure that all scientific articles pertaining to the occupational therapy workforce have been mapped. Furthermore, we have searched for key, mainstream research databases but not for others with a more regional focus such as the Latin American and Caribbean Health Sciences Literature, Scientific Electronic Library Online, African Index Medicus, etc. This fact may contribute to an underrepresentation of scientific literature published in journals merely indexed in regional databases. Additionally, we recognize that some studies have findings that could also pertain to more than one category. For example, findings on rural private practices [46] could have been framed either alongside studies on rural retention or studies on distribution by sector. Salary issues can also relate to either attraction, recruitment, and retention issues or be shaped by supply–demand imbalances; investigations in this area could therefore be part of other categories rather than standing as an independent category.

## 5. Conclusions

The occupational therapy workforce research worldwide does not provide discernable trends beyond some studies on the issues of attractiveness and retention in Australia and supply and demand in the USA. Many prevalent research topics or methodological approaches in health workforce research were not approached or rarely addressed, including investigations studying the context of LMICs, where often there are no education programs for occupational therapists. Additionally, programs of research (i.e., with an evolving sequence of studies) or continuous updates were nearly absent. Scattered in topics, timings, or contexts, and over-reliant on descriptive or exploratory methods, the occupational therapy research can be best described rather than thoroughly aggregated and synthesized toward generating a sound body of knowledge that can be both generalizable and context-sensitive—i.e., outlining what is common and what varies across health systems, countries, and geographies that benefit from tailored approaches. To address the identified gaps in content coverage or in the use of contemporary research approaches, there is a need for a global strategy on the strengthening of the occupational therapy workforce research worldwide. The WFOT plans to use the results of this scoping review to inform a consultation process on the directions for such a global strategy.

## Figures and Tables

**Figure 1 ijerph-19-05307-f001:**
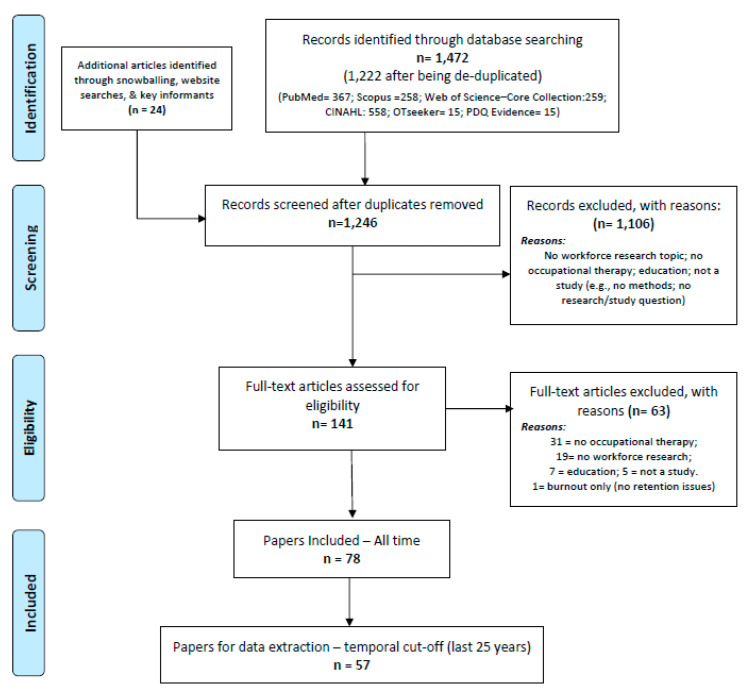
The Preferred Reporting Items for Systematic Reviews and Meta-Analyses (PRISMA) flowchart of this review.

**Table 1 ijerph-19-05307-t001:** Inclusion categories for the major topics of workforce research included, synthesized from the review protocol.

Inclusion Category	Category Type
1	Workforce supply (e.g., supply of practicing therapists or mapping their profile)
2	Workforce production (e.g., graduates supply or entry-level requirements)
3	Workforce needs, demands or supply-need/demand shortages; forecasts
4	Employment trends (e.g., (un)employment patterns, unfilled vacancies)
5	Workforce distribution (e.g., per geographies, practice area, public vs. private sectors)
6	Geographical mobility (e.g., (e/i)migration; internationally trained workers)
7	Attractiveness and retention (e.g., salaries, incentives, job satisfaction, intention to leave the profession, recruitment determinants)
8	Staff management and performance (e.g., human resources management, workload management, recruitment practices from managers, staffing and scheduling, burnout associated with performance or productivity)
9	Regulation and licensing (e.g., continuing education requirements, task-shifting, evaluating the impact of licensing or regulatory changes)
10	Systems-based or systematic analysis of workforce policies

## Data Availability

This study used published data.

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
