# Peer review of "Type of Findings Generated by the Occupational Therapy Workforce Research Worldwide: Scoping Review and Content Analysis"

_ijerph, 2022, doi:10.3390/ijerph19095307_

Round 1
Reviewer 1 Report
The authors present a scoping review focusing on knowledge produced about occupational therapy workforce. The topic is very welcome, once the synthesis in this field is necessary, or, in other words, it is necessary to describe its state of the art.
My first point to discuss here is the keyword applied to approach the topic. Although “workforce” is a recorded keyword on MeSh, for example, it is not present in all articles approaching this topic. Some of them present surveys of occupational therapists’ work, using “professional practice”, “staff”, and other keywords. Thinking of it, and at the same time recognizing it is not possible to achieve all articles about the topic, which is spread through different keywords; I suggest the authors point it as a limitation of the study, and consider the possibility of existence of other articles about the topic, including in the databases searched.
Starting the article, on line 33, the authors define occupational therapy as a “health professional”. However, on lines 414 to 417, based on the finds of the scooping review, they say: “Occupational therapists can work in several practice areas of the healthcare sector, in communities for health promotion or occupational justice, as well as in the educational, labor, vocational, and social sectors”. Consequently, I suggest reviewing the first definition to include all sectors where occupational therapists work to encompass all its characteristics worldwide.
On the “2. Material and Methods”, specifically on 2.1, I suggest exhibiting the keywords and the combination applied for screening the articles. Although the authors have already published the protocol of the review, it would be important to clarify here how the universe of the 57 articles was found.
Still on the Methods, item 2.2, the authors say only papers in English were verified (lines 88-89). However, there are bilingual articles (articles published in English and another language simultaneously), with the original in the other language. For example, the article cited in the Brazilian Journal of Occupational therapy is published in Portuguese and English languages, and the original is the Portuguese version. I suggest verifying another possible situation like that and specify it.
On item 4.1, Study Limitations, as I said earlier, I think it is important to approach the issue related to the search keywords. Another relevant point is related to databases. The authors did not search on others databases not situated in the more common countries, like the USA and UK. For example, LILACS (Latin America and Caribbean Health Science Literature) is an important database in the health field in Latin America, and SciELO (Scientific Electronic Library Online) covers 15 countries, including countries in Latin America, South Africa, Portugal, and Spain. Considering the proposition of the discussion around the world, it should highlight the limitations imposed by databases.
The conclusion is important, indicating the necessity for researchers to generate knowledge “that can be generalizable and context-sensitive” (lines 507-508). It would be great to have some more words about it, positioning the authors related to this and, if it is the case, advocating for this.
Congratulations for this article and I hope it can bring more discussion and indications of research development in the occupational therapy field.
Author Response
We submit an attached document with the responses to the each comment of each of three reviewers.

Reviewer 2 Report
A thorough review that is thoughtfully organised with clear cautions to reader on potential limitations given the nature of the review (e.g., studies not appraised).
the paper provides a helpful lay of the land in OT workforce research and communicates findings effectively.
It would be helpful to consider in the discussion potential approaches to future research given the findings of this scooting study.
Author Response
We submit an attached document with the responses to the each comment of each of three reviewers - the comments directly for this reviewer are under the "Reviewer 2" heading

Reviewer 3 Report
In general, a scoping review has a broad research question. I missed this question to inform what the researchers aim to investigate.
2.1. It is not clear the keywords you used.
Figure 1 is based on the Prisma flowchart and needs to be referenced.
3.1.1 Just summarise the study with no need to refer to p values.
Line 361- please correct punctuation: “[78]: This f…”.
Line 368- Instead of production isn’t better “training new professionals”?
Your study brings important data about the OT workforce, however, I missed a section “Implications” about what needs to be done? More guidance/guidelines of WFOT to support studies on the OT workforce? A community of practice to discuss and generate actions? It is sad to learn that in different countries the professionals intend to leave the profession. This can also be discussed under concepts of occupational injustice (for example, occupational imbalance).
Author Response
We submit an attached document with the responses to the each comment of each of three reviewers - the comments directly for this reviewer are under the "Reviewer 3" heading

Round 2
Reviewer 1 Report
The authors added elements suggested by the reviewers, clarifying the article and its characteristics. I suggest its publication.
I would like just to highlight my disagreement with the author’s response related to defining occupational therapy as a “health professional”. I understand and agree with the point presented by the authors: “Occupational therapists or other health professionals (e.g., nurses, medical doctors, physiotherapists) may all work in many other sectors and still are health professionals when they do so”. However, in a global perspective, as proposed by this article, occupational therapy is more than a health professional. Health sector is still the biggest place of working, with the biggest number of professionals, however we have others working in the education fields, social field, justice field and others, who should have been included in a global perspective, as proposed by this article. Hence, occupational therapy is as a health professional, including working in different sector services; and they are social and/or educational professionals, depending on the nature of the services and/or policies who they work in. Which means they are not health professionals in some sectors, but rather professionals promoting social protection, educational access, justice mediation etc. If the core of the profession is occupation/everyday life, it is not restricted to the health field. I prefer the broader words of the authors: “Occupational therapists can work in several practice areas of the healthcare sector, in communities for health promotion or occupational justice, as well as in the educational, labor, vocational, and social sectors”. This is an interesting point of discussion in the occupational therapy field.